# Progressive Gradient Flow for Robust N:M Sparsity Training in Transformers

Abhimanyu Rajeshkumar Bambhaniya[1*], Amir Yazdanbakhsh[2*], Suvinay Subramanian[3]
Sheng-Chun Kao[4], Shivani Agrawal[3], Utku Evci[3] , Tushar Krishna[1]
[1]Georgia Institute of Technology, [2]Google DeepMind, [3]Google, [4]Waymo

abambhaniya3@gatech.edu, ayazdan@google.com, suvinay@google.com, felixkao@google.com,
shivaniagrawal@google.com, evcu@google.com, tushar@ece.gatech.edu

N:M Structured sparsity has garnered significant interest as a result of relatively modest overhead and improved efficiency. Additionally, this form of sparsity holds considerable appeal for reducing the memory footprint owing to their modest representation overhead. There have been efforts to develop training recipes for N:M structured sparsity, they primarily focus on low-sparsity regions ($\sim$50%). Nonetheless, performance of models trained using these approaches tends to decline when confronted with high-sparsity regions (>80%). In this work, we study the effectiveness of existing sparse training recipes at *high-sparsity regions* and argue that these methods fail to sustain the model quality on par with low-sparsity regions. We demonstrate that the significant factor contributing to this disparity is the presence of elevated levels of induced noise in the gradient magnitudes. To mitigate this undesirable effect, we employ decay mechanisms to progressively restrict the flow of gradients towards pruned elements. Our approach improves the model quality by up to 2% and 5% in vision and language models at high sparsity regime, respectively. We also evaluate the trade-off between model accuracy and training compute cost in terms of FLOPs. At iso-training FLOPs, our method yields better performance compared to conventional sparse training recipes, exhibiting an accuracy improvement of up to 2%. We have open-sourced our verified implementation and it can be found at https://github.com/abhibambhaniya/progressive_gradient_flow_nm_sparsity.

## 1. Introduction

A prevailing tendency in state-of-the-art DNN is the rapid increase in their model [1–5]. To address the deployment challenges of these models, a large body of research proposes quantization [6–10], sparsification [11–20], and distillation [21]. This paper centers its attention on *sparsification/pruning* offering the following benefits: (a) improved performance [22], (b) reduce memory usage [23], & (c) higher energy efficiency [24, 25].

While appealing, sparsification predominantly revolves around the inherent trade-offs between the quality of the model and compression ratio[1]. For example, some studies [13, 26] have demonstrated promising results in achieving unstructured sparsity levels of around 90%-95% in image classification models, while maintaining the quality of dense models. Similarly, the noticeable achievements of transformer-based models, primarily driven by their exponential growth in model size [27], have stimulated interest [28–31] in exploring sparsification recipes for such models with high sparsity ratio. This serves as a significant incentive for the sparsification of attention-based models, as it enables the pruning of a substantial number of model parameters (>70%) [32, 33]. Despite its inherent ability to trim the memory footprint of large models, the realization of unstructured sparsity in hardware poses nontrivial challenges for acceleration. The sparsity-induced models frequently exhibit comparable or inferior performance to their dense counterparts because of the additional intricacies involved in compression/decompression of model parameters [34–39].

As such, structured sparsity has gained significant popularity because of its hardware-friendly characteristics. [16, 40–45] found that employing fine-grained N:M structured sparsity, has the potential to mitigate the

---

[*]Equal contributions.

[1]We designate algorithmic-wise factors such as accuracy, recall, and precision as *model quality*. and denote model runtime/latency as *model performance*.

Second Conference on Parsimony and Learning (CPAL 2025).

degradation in quality. Moreover, the debut of 2:4 structured-sparse tensor core in GPU Ampere architecture [34] has generated additional enthusiasm in developing efficient N:M training recipes. Although recent methods [22, 46–50] demonstrate acceptable quality, their main focus lies in addressing sparsity levels up to 2:8. These methods, however, less effective when dealing with high sparsity regimes such as 1:16, 1:32, and higher. Through our studies, we identify that elevated levels of induced noise in the gradient magnitudes constitute a notable contributing factor to such quality degradation. This phenomenon can be primarily attributed to either the absence [51, 52] or perturbation of gradient flow of existing sparse training recipes. Building on the insights our experiments, we introduce alternative training recipes that demonstrate substantial improvements in model quality, particularly at high sparsity regime. We made the following contributions:

- **The impact of gradient perturbations becomes increasingly evident at elevated levels of sparsity, leading to a deterioration in the quality of the model.** We present empirical evidence that SR-STE, a state-of-the-art N:M structured training recipe [22], is less effective at high sparsity regions, $> 75\%$. We attribute this to the nontrivial perturbation of gradient magnitudes. This perturbation during the initial stages of training[2] adversely amplifies the variance of gradients, resulting in a diminished model quality.
- **Gradient flow is all you need.** In order to alleviate the adverse effects of noisy gradients, we introduce a class of decaying-based sparse training recipes tailored for N:M structured sparsity. The fundamental principle underlying these methods involves progressively limiting the flow of gradients for *pruned weights*, while allowing the gradients to freely flow at the early stages of training. Our results demonstrate that the decaying-based methods consistently outperform SR-STE by up to 2%-5% in terms of model quality, while pruning $\sim 97\%$ of parameters.
- **Decaying-based sparse training recipes require less training FLOPs.** To better understand the computational overhead of the proposed sparse training recipes, we present the trade-off between model accuracy and training compute cost in term of FLOPS. The results show that at iso-quality, our method requires $> 30\%$ fewer training FLOPs compared to SR-STE.

## 2. Background and Related Works

This work focuses on weight sparsity, which poses a significant challenge in serving attention-based models.

### 2.1. Computation Flow of Sparse Training Recipes

Figure 1 summarizes the computation flows of various training recipes for the sparsification of weights. A sparsification recipe broadly entails 1) pruning criteria, 2) pruning schedule, and 3) sparsity pattern.

**(1) Pruning criteria.** The pruning criteria refers to the set of criteria used to determine the specific elements within the weight tensor that should be pruned. Magnitude pruning selects the pruning elements based on their absolute values, is one of the most widely used criteria for sparsification [12, 13, 17, 36, 53–56]. Recent work employs other criteria such as gradient [57, 58], Hessian [59], connection sensitivity [53], and importance estimation [60]. In this paper, we use magnitude pruning, following SR-STE [22] the state-of-the-art structured N:M training recipe.

**(2) Pruning schedule.** We classify the pruning schedules into the following broad categories:

- Fine-tuning with one-shot pruning$\rightarrow$ This approach [46, 47, 53, 54] involves training a dense model, followed by on-shot weight pruning. Subsequently the pruned model is fine-tuned to regain the lost quality.
- Fine-tuning with iterative pruning$\rightarrow$ This method [11–17, 38, 58, 61–63] trains a dense model followed by iterative cycles of pruning and re-training, which shows a greater capacity to regain lost quality.
- From-scratch with learned pruning pattern$\rightarrow$ This pruning recipe [11, 64] establishes the sparsity pattern based on pretrained dense model and subsequently trains a sparse model from scratch.
- From-scratch while learning sparsity pattern$\rightarrow$ This approach [55, 58, 65–69] trains a sparse model from scratch while concurrently learning the sparsity mask.

**(3) Sparsity pattern.** We broadly categorize sparsity patterns into following groups:

---

[2]Recent studies for dense models [49, 51] have shown that the early stage of training (critical region) is imperative in the quality of training recipes.

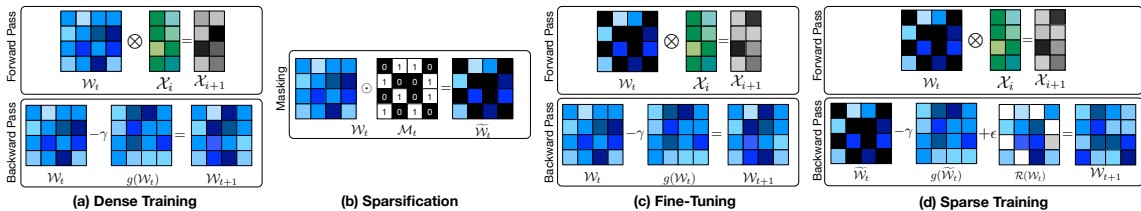

Fig. 1: The computation flow of (a) dense training, (b) sparsification, (c) fine-tuning, and (d) sparse training (e.g. SR-STE). $\widetilde{\mathcal{W}}$ represents a pruned matrix that is computed by element-wise multiplication ($\odot$) of $\mathcal{W}$ and its sparsification mask ($\mathcal{M}$). Sparse training recipes, such as SR-STE, introduce a "*sparse refining*" regularizer ($\mathcal{R}$) to adjust the gradient terms for pruned elements.

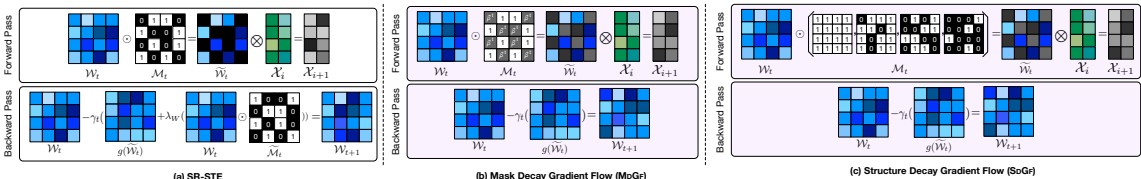

Fig. 2: An overview of different sparse training recipes (a) SR-STE [22], (b, c) proposed decaying mechanisms in this work. (b) indicates decaying binary mask values for pruned weights (MDGF), whereas (c) gradually change the N:M sparsity patters at different intervals (SDGF).

- Unstructured Sparsity refers to the process of pruning a model without imposing any constraints on the sparsity pattern [13, 36, 53–55]. This sparsity pattern is known to be able to prune the model size to an order of magnitude smaller while retaining a similar model quality as its dense counterpart at the cost of increased runtime overhead.

- Coarse-grained Structured Sparsity enforces coarse-grained sparsity patterns, including techniques like filter/channel pruning [14, 70, 71] and block-wise pruning [35, 62, 71, 72]. By skipping the entire computation of a tensor, this sparsity pattern often yields speedup in natively-dense accelerators such as GPUs and TPUs. Nevertheless, this trade-off often results in a reduction in model quality.

- Fine-grained Structured N:M Sparsity prunes (M-N) out of M consecutive elements. Several preliminary studies rely on special threading and grouping techniques [16] or specialized sparse accelerators [40] to exploit this fine-grained sparsity pattern. With the inclusion of 2:4 GEMM support in GPU Ampere architecture [34], recent work starts to investigate effective training recipes for N:M sparsity patterns to harness the existing accelerators [22, 46–48].

**Other related work.** Other work has also investigated N:M structured sparsity in attention-based models. Figure 2(a) demonstrates the weight update scheme for the forward and backward pass of SR-STE [22]. SparseGPT [50] introduces a post-training sparsification recipe tailored for GPT-family models. SparseGPT shows on-par model quality with up to 50% weight pruning under unstructured and N:M structured sparsity. Finally, selective weight decay (SWD) [73] is a pruning method based on Lagrangian smoothing, which penalizes weights that are selected for pruning. However, SWD neither explores attention models nor provides training recipes for N:M structured sparsity.

## 3. Decaying-based Sparse Training Recipes

This section covers the class of decaying-based training recipes for fine-grained N:M sparsity. The main premise of these recipes is to allow the gradient to flow through weight tensors in a controlled way to prevent induced noise in the gradients. We broadly classify the proposed decaying-based training recipes into: (a) "**M**ask **D**ecay **G**radient **F**low" (MDGF) and (b) "**S**tructure **D**ecay **G**radient **F**low" (SDGF), each with sub-variants which we discuss in details below. In contrast to [22], we intentionally refrain from modifying the gradient update rules in either of these categories. Instead, we use different update rules for sparsity pattern or sparsity mask tensor, facilitating unimpeded gradient flow during the entire sparse training process.

**Implementation.** In order to implement these methods, we employ the process of pruning dense weight tensors ($\mathcal{W}_t$) to generate sparse weight tensors ($\widetilde{\mathcal{W}_t}$), adhering to the following rule during the forward pass:

$$\widetilde{\mathcal{W}} = \mathcal{F}(\mathcal{W}, N, M, \Phi, \beta, j)$$
$$= \mathcal{W} \odot [\Phi(\mathcal{W}, N, M, j) + \mathcal{D}(j)(1 - \Phi(\mathcal{W}, N, M, j))]$$

Here $\odot$ represents the Hadamard product. $\Phi(\cdot)$ and $\mathcal{D}(\cdot)$ calculate a decaying-based binary mask and decay mask factor, respectively. ($j$) denotes the training step count. Each function's implementations establish distinct decaying-based training recipes. $\Phi(\cdot)$ calculates a binary mask that matches the dimensions of the input weight tensor ($\mathcal{W}$). The location of 0s and 1s in the binary mask refers to pruned and unpruned weights, respectively. In fine-grained N:M structured sparsity with magnitude pruning, $\Phi(\cdot)$ assigns a value of 1 to the N weight tensor elements with the highest absolute magnitude within a contiguous block of M elements. Simultaneously, it enforces all the other elements with the block to be set to 0. In addition, $\mathcal{D}(\cdot)$ calculates the decaying factor for binary mask according to the target decaying-based training recipe. It should be noted that ($[\Phi(\mathcal{W}, N, M, j) + \mathcal{D}(j)(1 - \Phi(\mathcal{W}, N, M, j))]$) is not a sparse matrix during intermediate steps. However, as $\mathcal{D}(j)$ decays to 0 over the course of training, ($\widetilde{\mathcal{W}}$) ultimately equals ($\mathcal{W} \cdot \Phi(\mathcal{W}, N, M, j)$), which is sparse.

❶ **Mask Decay Gradient Flow (MDGF).** In the first training recipe Figure 2 (b), we propose the use of a diminishing value ranging from 1 to 0, as opposed to the commonly-used binary pruning mask (e.g., "0" → pruned and "1" → dense). Note that for the mask-decay training recipes the function $\Phi(\cdot)$ produces a mask tensor either with all ones (dense training) or with a sparsity pattern following target N:M fine-grained structured sparsity. In the initial epochs, we use a mask comprising solely of ones and assign a constant value of 1 to $\mathcal{D}(\cdot)$, i.e., dense training.

Upon staring sparse training phase, $\mathcal{D}(\cdot)$ produces gradually diminishing floating-point values between 1 and 0. The output of function $\mathcal{D}(\cdot)$ depends on current decaying interval. Using a diminishing decaying factor enables gradient flow for both pruned and unpruned weights. This is in contrast to prior work in which $\mathcal{D}(\cdot)$ is null which may cause instability in the training process. We propose two new implementations for $\mathcal{D}(\cdot)$:

- MDGF-Linear uses $\mathcal{D}(j) = max(1 - \beta_\tau \times j, 0)$ that reduces the decay mask values linearly with respect to training steps.
- MDGF-Exponential, as its name implies, we use $\mathcal{D}(j) = e^{-\beta_\eta \times j}$, indicating an exponential decrease in the mask decay value relative to the ongoing training step.

The value of $\beta_{\tau/\eta}$ determines the rate of decay. To ensure a binary mask value for the target N:M sparsity pattern, after sufficient decaying intervals, $\mathcal{D}(\cdot)$ approaches zero. After reaching the target N:M sparsity pattern, we proceed with few additional training epochs to restore the model accuracy. We postulate that using non-binary pruning mask values facilitates the smooth propagation of gradients in pruned weights, resulting in more stable sparse training. For practical use, we recommend setting the decay rate such that the decay factor reaches zero when approximately 70% of the training budget is completed, allowing for sufficient fine-tuning of the final sparse weights.

❷ **Structure Decay Gradient Flow (SDGF).** SDGF decays the structure of the pruning mask, e.g. gradually altering the sparsity level, e.g. $\frac{3}{4} \mapsto \cdots \mapsto \frac{1}{4}$. In contrast to MDGF, this method strictly confines the pruning mask values to either 1 or 0, e.g. $\mathcal{D}(\cdot) = 0$. We propose two alternative implementations of $\Phi(\cdot)$, (a) *Stepwise* and (b) *Geometric*.

The SDGF-*Stepwise* starts by inducing M-1:M structured sparsity. Subsequently, it gradually increase the level of sparsity following $\frac{M}{2^d} : M$ formulation in which $d$ denotes the index of the decaying interval, until $\frac{M}{2^d} == N$. For example, to retain a target sparsity level of 1:8, the method applies the following sparsity patterns at different decaying interval $\frac{7}{8} \mapsto \frac{4}{8} \mapsto \frac{2}{8} \mapsto \frac{1}{8}$.

The core idea of SDGF-*Geometric* is to maintain a constant ratio of $\frac{N}{M}$ throughout successive decay intervals by adjusting the values of N and M in proportion to each other. In all experiments, we configure $\Phi(\cdot)$ to be $\frac{k \times M}{2^d} : \frac{k \times N}{2^d}$. The value of $k$ is set to 16, unless specifies otherwise. We empirically find that $k > 16$ offers negligible improvements in terms of model quality. For example, for a target sparsity of 1:8, we induce the following sparsity patterns at each decaying interval, $\frac{16}{128} \mapsto \frac{8}{64} \mapsto \frac{4}{32} \mapsto \frac{2}{16} \mapsto \frac{1}{8}$. For both recipes, we evenly partition the total sparsification epochs throughout the decaying intervals. Fundamentally, this approach

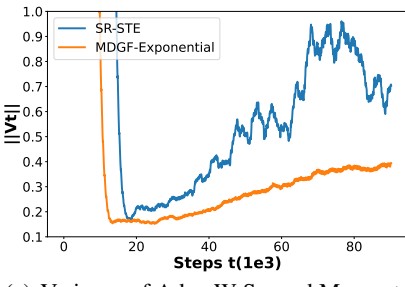 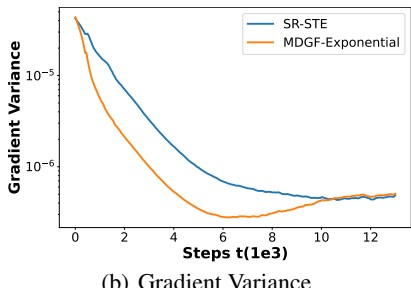

(a) Variance of AdamW Second Moment

(b) Gradient Variance

Fig. 3: Trends for different indicators of gradient values during training. Data from ViT-tiny trained on CIFAR-10 with 1:16 sparsity pattern. (a) and (b) show the running average of the variance of AdamW second moment and gradient variance, respectively.

follows a hypothesis akin to MDGF. Enabling the flow of gradients of pruned weights throughout the model potentially leads to higher model accuracy.

# 4. Impact of Gradient Flow in Sparsification

To better understand the impact of gradient flow while sparifying the weights, we follow the insight that when the decaying variance of the noisy gradient is large, the algorithm might spend much time bouncing around, leading to slower convergence and worse performance [52]. Through MDGF and SDGF, we allow a smoother and more stable gradient flow during backpropagation compared to SR-STE, thereby reducing the noise introduced by high-sparsity constraints.

In order to observe the effect of proposed decay methods, we conducted an empirical analysis to compare the gradient values of MDGF-Exponential and SR-STE [22]. We created a compact version of ViT with three encoder layers, each with three attention heads, and an embedding size of 192. We trained this model on CIFAR-10 [74] for 200 epochs with batch size 64 with AdamW optimizer. To understand the impact of sparsification, we collect and analyze two different metrics, namely *second moment* and *gradient variance*. These values are an indication of how effective the gradient estimations are for training [75–77].

## 4.1. Analysis of Second Moment Estimates

3(a) shows the variance of the second moment term (exponential moving average of squared gradient values) for Feed-Forward (FF) layers in the model. We observe that in MDGF, the variance steadily decreases in magnitude, whereas in SR-STE, the variance stays at the relatively high level even at the later stages of training. Prior study [75–77] correlate lower variance of the second moment with faster convergence rate during training and better model accuracy. This suggests that the gradient noise induced by SR-STE have negative impact on the convergence of the model and model accuracy.

## 4.2. Analysis of Gradient Noise

Figure 3(b) shows the variance of absolute back-propagation gradients. These values can be interpreted as the amount of noise in the gradient estimates. Similar to the previous study, we collect the gradients of Feed-Forward(FF) layer in tiny-ViT. We observe that in MDGF, the variance of gradients decreases quickly, whereas in SR-STE, the variance of gradients has a lower slope (e.g. taking a larger number of steps). When the variance of the gradient is higher, the optimizer spends time bouncing around, leading to slower convergence and lower performance [51, 52]. The variance for MDGF-exponential comes down rather quickly thus the gradients are less noisy compared to SR-STE. This would result in higher accuracy for MDGF-Exponential. When observing the final validation accuracy of the two runs, we confirm our intuitive conclusions as the SR-STE accuracy is lower compared to MDGF-Exponential accuracy.

Table 1: The compute and memory contributions of the three major layers in Transformers. These estimations are made for ViT-Base. The FF layers account for around 64% of overall FLOPs and 66.6% of parameters. We use sequence length 196 to read image of 224x224.

|  | Einsum (Logit & Attend) | Projections (Q/K/V/O) | Feed Forward (FF1/FF2) |
|---|---|---|---|
| (T)FLOPS | 1.42 (4%) | 11.1 (32%) | 22.20 (64%) |
| Params (MB) | 0.0 (0%) | 28.31 (33.3%) | 56.62 (66.6%) |

# 5. Experiment

In this section, we evaluate the effectiveness of various training recipes for N:M fine-grained structured sparsity in a range of attention-based models and tasks, such as image classification, language translation and understanding. Motivated by the relatively substantial contribution of FF layers (Table 1) in total FLOPs

Table 2: ImageNet-1K Top-1 validation accuracy on `ViT-Base` across different N:M sparsity patterns and training recipes.

| Sparse Target | Dense | SR-STE | MDGF-Linear | MDGF-Exponential | SDGF-Stepwise | SDGF-Geometric |
|---|---|---|---|---|---|---|
| 2:4 (FF) | 76.389 | **77.761** | 77.613 | 76.381 | 77.081 | 77.363 |
| 1:4 (FF) | 76.389 | **78.782** | 78.512 | 78.579 | 77.357 | 78.347 |
| 1:8 (FF) | 76.389 | 77.869 | 78.019 | 78.009 | 77.025 | **78.175** |
| 1:16 (FF) | 76.389 | 75.637 | 76.594 | **77.325** | 75.923 | 76.869 |
| 1:32 (FF) | 76.389 | 73.056 | 75.807 | **76.068** | 74.394 | 74.910 |
| 1:128 (FF) | 76.389 | 72.069 | 74.012 | **74.180** | 71.725 | 69.801 |
| 1:4 (FF) + 1:4 (QK) | 76.389 | 78.145 | 77.755 | 78.113 | 77.163 | **78.229** |
| 1:8 (FF) + 1:8 (QK) | 76.389 | 75.527 | 76.473 | **77.349** | 76.617 | 76.334 |
| 1:8 (FF) + 1:4 (QK) | 76.389 | 78.144 | 78.025 | **78.273** | 77.163 | 76.839 |
| 1:8 (FF) + 1:4 (QKV) | 76.389 | 78.222 | 78.319 | **78.319** | 77.309 | 78.213 |

($\sim 64\%$) and parameter count ($\sim 66.6\%$), we center our experiments around sparsification of these layers within the encoder and decoder blocks. In addition, we conduct experiments on the pruning of projection layers ($Q/K/V$) for a variant of `ViT-Base` [78], a variant of `SwinV2-Base` [79], and `T5X-Base` [1]. For `ViT-Base`, we use fixed-size patches (resolution $16 \times 16$) on images with resolution 224. In `SwinV2-Base`, we employ window sizes of $8 \times 8$ on images with resolution 256. For image classification tasks, we branched (commit: 1304589) our implementation from PyTorch Image Models [80] and use NVIDIA A100 GPUs for training on ImageNet-1K dataset [81]. For `T5X-Base`, we extend the official Google T5X release (commit: d3d3cbf) with sparsification training recipes and use Google TPUv3. We train these models from scratch using different training recipes across different patterns of N:M fine-grained structured sparsity. SR-STE [22] serves as the baseline sparse training recipe to assess the effectiveness of the proposed training recipes in terms of model accuracy. Appendix C have details about training hyperparameters, dataset details, and evaluation metrics.

## 5.1. Image Classification ↦ `ViT-Base` and `SwinV2`

**`ViT-Base` model quality.** Table 2 presents Top-1 validation accuracy for variations of N:M sparsity in `ViT-Base`, with the highest accuracy model indicated in bold. The "*Sparse Target*" column signifies the intended level of N:M sparsity. For example, a sparsity target of 1:32 indicates that sparse tensors exhibit at most one non-zero for every 32 contiguous elements. In low sparsity scenarios (e.g., 2:4 and 1:4), both MDGF and SR-STE demonstrate comparable performance. Nevertheless, with increases in either sparsity degree (e.g., 1:8 and higher) or the number of sparse layers, e.g., 1:4 ($FF$) + 1:4 ($QK$), employing SR-STE is detrimental to model quality. In contrast, the proposed decaying-based training recipes, MDGF and SDGF, yield the highest accuracy.

Interestingly, when aiming for a sparsity target of 1:32 (approximately 97%), MDGF-Exponential showcases a mere 0.3% reduction in accuracy compared to a fully dense model (76.389 vs. 76.068). Additionally, we notice that the model accuracy increases at modest sparsity degrees, specifically in 2:4/1:4/1:8 (FF) patterns, resulting in an improvement of up to $\Delta(Acc) = +2.4\%$ in 1:4 (FF). The increase in model accuracy, demonstrated in 4(a), can be attributed to Occam's Hill, wherein the positive impact of sparsity as a means of regularization is elucidated [82, 83]. The performance of MDGF-Exponential training recipe is comparable to that of SR-STE

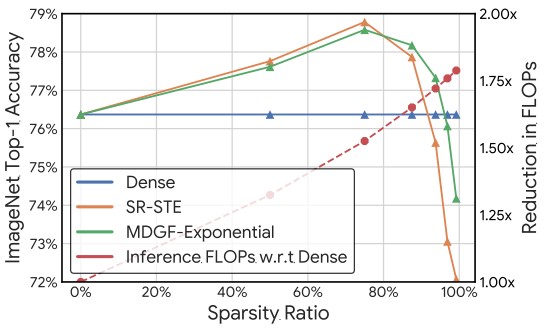

(a) Accuracy vs. Sparsity ratio showing Occam's hill.

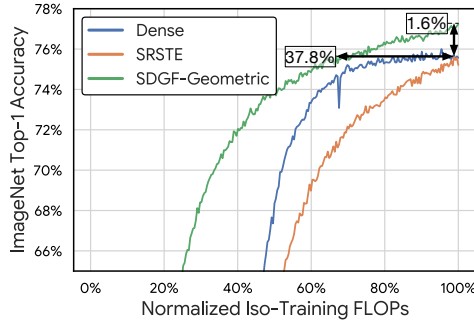

(b) Accuracy vs % of training epochs.

Fig. 4: `ViT-Base` trained on ImageNet-1K with different sparsity patterns and targets. (a) shows the Occam's hill where sparsity improves the model accuracy. The dashed red line shows the reduction in inference FLOPs at different sparsity ration. At high sparsity regime (>80%) MDGF yields better accuracy than SR-STE and (b) demonstrates model accuracy across training recipes (dense and sparse) at different training FLOPs. The vertical line indicates the proposed decaying method is better (1.6%) than dense model at given training FLOPS. The vertical line shows that the decaying based method reaches to dense model accuracy at 37.8% less training FLOPs.

in low-sparsity scenarios. However, the proposed MDGF-Exponential recipe far surpasses SR-STE when confronted with high-sparsity patterns.

As commercially available accelerator can not support high-sparsity patterns. In order to assess the *potential* performance benefits by comparing the savings in inference FLOPs as well as memory usage. Figure 5 visualizes the trade-off between accuracy and inference FLOPs across range of sparsity configurations and recipes. The results show that MDGF-Exponential with sparsity 1:16 provides similar accuracy as SR-STE 2:4 with 60% fewer inference FLOPs and 30% fewer parameters. Appendix D provides the details of FLOPs calculations.

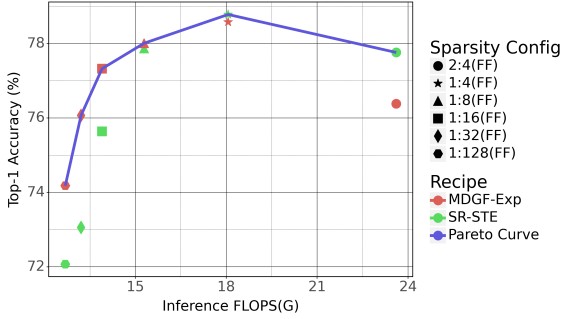

Fig. 5: FLOP vs. Accuracy for ViT-Base+ImageNet-1K.

**`SwinV2-Base` model quality.**

Table 3 demonstrate Top-1 validation accuracy for `SwinV2-Base`. Similar to `ViT-Base`, we observe that the decaying-based algorithms outperforms SR-STE across various N:M sparsity patterns. In 1:4 and 1:8 ($\mathcal{FF}$), SDGF-Geometric yields the highest Top-1 validation accuracy. Whereas, in high-sparsity patterns, MDGF-Exponential demonstrates superior performance compared to SR-STE. To summarize, the results from the two image classification models demonstrate that the proposed training recipes, MDGF and SDGF, which incorporate decaying-based approaches for N:M fine-grained structured sparsity, yield superior performance compared to SR-STE.

Table 3: ImageNet-1K Top-1 validation accuracy on `SwinV2-Base` across different N:M sparse patterns and training recipes.

| Sparse Target | Dense | SR-STE | MDGF-Exponential | SDGF-Stepwise | SDGF-Geometric |
|---|---|---|---|---|---|
| 1:4 (FF) | 83.45 | 82.355 | **82.491** | 82.267 | 82.469 |
| 1:8 (FF) | 83.45 | 81.437 | **81.466** | 81.382 | 81.382 |
| 1:16 (FF) | 83.45 | 80.154 | **80.542** | 80.386 | 80.274 |
| 1:32 (FF) | 83.45 | 78.972 | **79.545** | 76.480 | 79.277 |
| 1:8 (FF) + 1:8(QK) | 83.45 | 81.441 | **81.550** | 81.218 | 81.438 |

## 5.2. Language Understanding ↦ `T5X-Base`

We also analyze the efficacy of the proposed decaying-based training recipes for the language understanding task. We employ a dense pre-trained `T5X-Base` model trained on the C4 dataset with a span-corruption

objective [1]. The dense pre-trained model undergoes fine-tuning using the GLUE dataset [84] with various training recipes for N:M structured sparsity. Table 4 depicts the overall score, summarized across eight different GLUE tasks. We observer a consistent trend where SDGF outperforms SR-STE at high-sparsity patterns and increasing number of sparse layers. Notably, we observe a relative difference of $\Delta = +5.3$ in 1:8 ($\mathcal{FF}$) + 1:8 ($\mathcal{QKV}$) sparsity pattern. Appendix A and Appendix B provide details about the T5X-Base model, per-task evaluation metrics, and additional ablation studies.

Table 4: The GLUE overall score on the sparsified T5X-Base model across different N:M sparse training recipes and patterns.

| Model | Sparsity Target | Dense | SR-STE | SDGF-Stepwise | SDGF-Geometric |
|---|---|---|---|---|---|
| T5X-Base | 1:4 (FF) | 86.2 | **84.1** | 83.7 ($\Delta = -0.4$) | 83.4 |
| T5X-Base | 1:32 (FF) | 86.2 | 79.4 | **80.9 ($\Delta = +1.5$)** | 79.3 |
| T5X-Base | 1:8 (FF) + 1:8 (QK) | 86.2 | 75.8 | **80.7 ($\Delta = +4.9$)** | 76.8 |
| T5X-Base | 1:8 (FF) + 1:4(QKV) | 86.2 | 78 | **80.3 ($\Delta = +2.3$)** | 78.9 |
| T5X-Base | 1:8 (FF) + 1:8 (QKV) | 86.2 | 74.2 | **79.5 ($\Delta = +5.3$)** | 75.8 |

## 5.3. Language Translation ↦ Enc-Dec

Table 5: The translation accuracy on WMT task across different N:M sparsity patterns and training recipes.

| Model | Sparsity Target | Dense | SR-STE | SDGF-Stepwise | MDGF-Exponential |
|---|---|---|---|---|---|
| Enc-Dec (WMT) | 1:16 | 0.747 | 0.709 | **0.717** | **0.717** |
| Enc-Dec (WMT) | 1:32 | 0.747 | 0.707 | 0.713 | **0.714** |
| Enc-Dec (WMT) | 1:64 | 0.747 | 0.707 | 0.710 | **0.711** |
| Enc-Dec (WMT) | 1:128 | 0.747 | 0.707 | 0.708 | **0.711** |

Finally, we compare the performance of different sparse training recipes on WMT language translation task [85]. For that, we use an encoder-decoder transformer-based model [86] each with six layers and 16 heads, which is relatively smaller than T5X-Base. outlines the details about this model and the training hyperparameters.

Table 5 demonstrates the accuracy results across range of sparsity patterns and training recipes. We observe that SDGF and MDGF collectively outperform SR-STE across various N:M structured sparsity patterns. However, we note that the difference in accuracy achieved through different training recipes is relatively smaller. This can be attributed to the model size (6 layers vs. 12 layers in T5X-Base), as well as the nature of the translation task, which appears to be less sensitive to sparsity patterns and training recipes[3].

## 5.4. Recipe impact for CNNs.

While the primary focus of this work is on evaluating sparse training recipe for transformer models, for the sake of completeness, we also test the efficacy of our recipe on CNNs. We train ResNet-50 following two sparse training recipes (SR-STE and MDGF-Exponential) and across different sparse patterns (2:8, 1:8). We pruned all the convolution layers and evaluate Top-1 validation accuracy on CIFAR-10. Table 6 shows a similar pattern, decaying-based sparse training recipes outperform SR-STE in both cases.

Table 6: ResNet-50 Top-1 validation accuracy.

| Sparse Target | Dense | SR-STE | MDGF-Exponential |
|---|---|---|---|
| 2:8 | 85.09 | 83.33 | **83.60** |
| 1:8 | 85.09 | 80.78 | **82.48** |

## 5.5. Baseline Comparison

Table 7: Comparing various sparsification techniques by fine-tuning T5X on GLUE dataset.

| Sparse Target | SR-STE [22] | SNIP [53] | IDP [87] | MDGF-Exponential |
|---|---|---|---|---|
| 1:32 (FF) | 79.4 | 79.5 | 80.6 | **80.9** |

SR-STE is our primary baseline in our evaluations as it has shown good results in low-sparsity regions [2:4,1:4] and is considered SOTA for N:M training. We also compared against other techniques like Inherited Dynamic

[3]Model accuracy is less affected as we increase the sparsity level beyond 1:32.

Pruning (IDP) [87], and SNIP: Single-shot Network Pruning [53]. Table 7 compares the results on T5X with GLUE dataset. We also tried to test against LBC [88] but could not recreate the results shown in the paper.[4]

## 5.6. Inference speedup on Real Hardware.

Current SOTA hardware accelerates sparsity at a 2:4 ratio. However, adopting higher sparsity levels can enable even faster inference, even without dedicated hardware support for such patterns. This acceleration primarily arises from reduced memory movement in memory-bound kernels.

To evaluate end-to-end acceleration on GPUs, we measure the runtime of the *ViT-Large* [78] inference stage across various N:M sparse patterns, using the dense implementation as a baseline. We induce N:M sparse patterns only in the FF layers. Our results demonstrate substantial benefits from N:M sparsity, even for patterns beyond 2:4, during end-to-end model inference. Table 8 presents the speedups achieved on V100 and A100 GPUs for different N:M sparsity patterns compared to dense ViT implementations. It is important to emphasize that these gains primarily stem from reduced memory movement and optimizations via custom cuSPARSE APIs. Furthermore, we report standalone acceleration for sparse FFN kernels in Appendix E.

Table 8: Speedup of ViT Inference with various sparsity amounts in FF layers.

| Hardware | Dense | 2:4 | 1:4 | 1:8 | 1:16 | 1:32 | 1:128 |
|---|---|---|---|---|---|---|---|
| V100 | 1.0 | 1.542 | 2.202 | 3.019 | 3.460 | 3.558 | 3.380 |
| A100 | 1.0 | 1.953 | 2.614 | 2.958 | 3.014 | 3.129 | 3.259 |

# 6. Limitations and Future Works

This work explores effective high-ratio sparsity for self-attention models. While we evaluate MDGF and SDGF in isolation, combining them across training regions may yield better model quality. Our key finding is that high sparsity degrades gradient estimation, which we mitigate by progressively tightening gradient flow—a simple yet effective strategy across various models and datasets.

However, our approach has limitations. First, while empirically effective, it lacks a strong theoretical foundation explaining why gradient decay alleviates sparsity effects. Future work could provide a rigorous analysis. Second, our method focuses on structured sparsity at training time and may not extend to dynamic scenarios like KV cache compression in autoregressive models. Adapting it to such contexts remains an open challenge. Further, applying our techniques to autoregressive models, which suffer from high inference costs and memory constraints, is a promising direction. A deeper evaluation in this setting could enhance their impact and utility.

# 7. Conclusion

We study the efficacy of recent sparsity recipes for N:M sparsity across transformer-based models and find that conventional methods introduce significant gradient noise at high sparsity (>75%). To address this, we propose decaying-based training recipes, with MDGF-Exponential achieving state-of-the-art accuracy—improving vision models by  2% and language models by  5% at high sparsity. Our results highlight the critical role of gradient flow, especially in early training. For same training FLOPs, our approach improves accuracy by 2%. Additionally, MDGF-Exponential (1:16) matches SR-STE (2:4) accuracy while reducing inference FLOPs by  60% and parameters by  30%. Finally, real hardware tests show up to 3.38× speedup over dense implementations. The source code is open sourced and available in github.

# 8. Acknoledgement

We are deeply grateful to Cliff Young and Vincent Vanhoucke for their valuable feedback and thoughtful review of this paper. We further recognize the extended team at Google DeepMind, who enabled and supported this research direction. We thank numerous anonymous reviewers for their feedback which helped in making this work stronger. This work was supported in part by CoCoSys, one of seven centers in JUMP 2.0, a Semiconductor Research Corporation (SRC) program sponsored by DARPA.

---

[4]We have contacted the authors but cannot solve the issue.

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

# A. Ablations Studies

This section shows the various ablation studies we performed during our experiments.

## A.1. Effect of dense training steps ($d$)

Both our proposed methods, MDGF and SDGF include a dense training phase. We do an ablation study on different amounts of dense training steps(% of total steps) in Table 9. We perform this study on the language translation model (more implementation details in section §C.2.4) trained on WMT-17. We found that changing the dense step between 1.25% - 10% of the total training steps does not observably change the accuracy performance. However, empirically, we found that the dense training phase is still essential. The model cannot achieve as competitive accuracy without few epochs of dense training.

Table 9: Ablation: The effect of number of dense training steps ($d$).

| Accuracy | | MDGF-Linear | | | | SDGF-Stepwise | | | |
|---|---|---|---|---|---|---|---|---|---|
| Sparsity Target | | 1:16 | 1:32 | 1:64 | 1:128 | 1:16 | 1:32 | 1:64 | 1:128 |
| Dense steps (d) | 1.25% | 0.7155 | 0.7134 | 0.7106 | 0.7100 | 0.7157 | 0.7134 | 0.7108 | 0.7106 |
| | 2.5% | **0.7160** | 0.7127 | **0.7110** | 0.7093 | 0.7160 | 0.7136 | **0.7117** | 0.7100 |
| | 5% | 0.7157 | **0.7137** | 0.7103 | 0.7094 | 0.7164 | **0.7141** | 0.7107 | 0.7098 |
| | 10% | 0.7156 | 0.7126 | 0.7107 | **0.7104** | **0.7165** | 0.7128 | 0.7115 | **0.7107** |

## A.2. Effects of fine-tuning steps ($s$)

We also have a sets of study on number of fine-tuning steps in Table 10. We perform this study on the language translation model (more implementation details in section §C.2.4) trained on WMT-17. We found that for all of our proposed methods, the fine-tuning steps between 10% - 20% of the total training steps do not observably change the accuracy performance. However, empirically, we also found few steps of fine-tuning at the end are essential to recovering the accuracy.

Table 10: Ablation: The effect of number of fine-tuning steps ($s$).

| Accuracy | | MDGF-Linear | | | | SDGF-Stepwise | | | |
|---|---|---|---|---|---|---|---|---|---|
| Sparsity Target | | 1:16 | 1:32 | 1:64 | 1:128 | 1:16 | 1:32 | 1:64 | 1:128 |
| Fine-tuning steps (s) | 10% | 0.7153 | 0.7130 | **0.7107** | **0.7098** | **0.7160** | **0.7125** | **0.7095** | **0.7072** |
| | 20% | **0.7161** | **0.7132** | 0.7106 | 0.7097 | 0.7121 | 0.7093 | 0.7081 | 0.7065 |

## A.3. Effect of ($\beta^t$) in MDGF-Linear

We also study on effect of decay rate on model's accuracy in Table 11. We do experiments with varying $\beta^t$ for ViT-Base trained on Imagenet-1k for different sparsity targets.

We observe that a higher decay rate is beneficial at low sparsity targets (2:4,1:4), but for targets higher than 1:8, we found lower decay rate works better.

Table 11: Ablation: The effect of mask decay rate ($\beta^t$) for MDGF-Linear.

| Sparsity Target | | 2:4 | 1:4 | 1:8 |
|---|---|---|---|---|
| Mask decay rate ($\beta^t$) | 0.0002 | 77.495 | 78.448 | **78.019** |
| | 0.001 | **77.613** | **78.512** | 76.4075 |

# B. Detailed Results for T5X-Base Sparsification on GLUE Dataset

We compared sparsification methods N:M block sparsification against state-of-the-art technique, SR-STE on. T5 model uses a span-based masked language modeling (MLM) objective. T5 models were introduced in [1] and the updated models are available at T5X-github. We train a pre trained t5x-base model on GLUE dataset [84].

The main paper shows a snapshot of the performance across various sparsity targets using the overall score as metric. Table 12 presents all 9 scores for each sparsification technique and sparsity target.

Table 12: GLUE full score using various T5X-base with different N:M sparse targets and various sparsification techniques.

| | | overall score | CoLA | MNLI matched | MNLI mismatched | MRPC | QNLI | QQP | RTE | SST-2 | STS-B |
|---|---|---|---|---|---|---|---|---|---|---|---|
| Dense | - | 86.2 | 58.9 | 87.2 | 87 | 92.4 / 89.2 (90.8) | 93.6 | 92.0 / 89.2 (90.6) | 82.3 | 95 | 90.1 / 90.0 (90.0) |
| SR-STE (Zero Dense) | 1:4 | 83.1 | 41.8 | 85.2 | 85.3 | **92.8 / 90.0 (91.4)** | 92.3 | 91.8 / 88.9 (90.3) | 79.1 | **93.6** | **89.5 / 89.2 (89.3)** |
| SR-STE (10K Dense) | 1:4 | **84.1** | 48.1 | **85.7** | **85.6** | 92.4 / 89.5 (91.0) | 92.1 | **91.8 / 89.0 (90.4)** | **82.7** | **93.6** | 87.9 / 87.7 (87.8) |
| MDGF-Stepwise (10K Dense) | 1:4 | 83.7 | **48.8** | 85.3 | 85.4 | 92.4 / 89.2 (90.8) | 92.3 | 91.8 / 89.0 (90.4) | 80.5 | 93.5 | 86.5 / 86.3 (86.4) |
| MDGF-Geometric (Zero Dense) | 1:4 | 83.3 | 48.4 | 85.3 | 85.3 | 92.0 / 89.0 (90.5) | 91.8 | 91.8 / 88.9 (90.3) | 78 | 92.8 | 87.3 / 87.4 (87.3) |
| MDGF-Geometric (10K Dense) | 1:4 | 83.4 | 47.2 | 85.4 | 85.3 | 92.6 / 89.7 (91.1) | 92 | **91.8 / 89.0 (90.4)** | 79.8 | 92.9 | 86.7 / 86.4 (86.5) |
| SR-STE (Zero Dense) | 1:32 | 77.1 | 19 | 81.3 | 81.3 | 90.9 / 87.0 (89.0) | 86.9 | 90.6 / 87.4 (89.0) | 71.1 | 89.9 | 86.7 / 86.8 (86.8) |
| SR-STE (10K Dense) | 1:32 | 79.4 | 29.4 | 82.2 | 82.6 | 91.5 / 88.5 (90.0) | 89.6 | 91.2 / 88.2 (89.7) | 72.6 | **91.4** | **87.1 / 87.2 (87.2)** |
| MDGF-Stepwise (10K Dense) | 1:32 | **80.9** | **38.3** | **83.6** | **83.7** | **92.5 / 89.7 (91.1)** | 90.5 | **91.5 / 88.5 (90.0)** | **74.4** | 91.2 | 85.2 / 85.0 (85.1) |
| MDGF-Geometric (Zero Dense) | 1:32 | 77.6 | 20.2 | 81.3 | 81.6 | 90.8 / 87.7 (89.2) | 87.2 | 90.8 / 87.7 (89.2) | 73.3 | 90.1 | 85.8 / 85.5 (85.6) |
| MDGF-Geometric (10K Dense) | 1:32 | 79.3 | 29.2 | 82.3 | 82.9 | 91.3 / 88.0 (89.6) | 90.4 | 91.3 / 88.3 (89.8) | 73.3 | 90.5 | 85.4 / 85.4 (85.4) |
| SR-STE (Zero Dense) | 1:8(FF) + 1:8(QK) | 74.4 | 15.7 | 77.2 | 77.6 | 89.9 / 85.8 (87.8) | 83.6 | 89.7 / 86.2 (87.9) | 67.5 | 88.2 | 84.1 / 83.9 (84.0) |
| SR-STE (10K Dense) | 1:8(FF) + 1:8(QK) | 75.8 | 19.9 | 78.6 | 79.4 | 89.7 / 86.0 (87.9) | 84 | 90.1 / 86.7 (88.4) | 70 | 89.4 | 84.5 / 84.2 (84.4) |
| MDGF-Stepwise (10K Dense) | 1:8(FF) + 1:8(QK) | **80.7** | **38.7** | **83.1** | **83.2** | **90.9 / 87.7 (89.3)** | **89.9** | **91.2 / 88.2 (89.7)** | **76.2** | **91.9** | **84.5 / 84.5 (84.5)** |
| MDGF-Geometric (Zero Dense) | 1:8(FF) + 1:8(QK) | 75.8 | 21.6 | 78.8 | 79 | 90.0 / 86.0 (88.0) | 83.6 | 90.1 / 86.6 (88.3) | 69.7 | 88.9 | 84.0 / 83.9 (83.9) |
| MDGF-Geometric (10K Dense) | 1:8(FF) + 1:8(QK) | 76.8 | 22.3 | 80.7 | 80.9 | 89.8 / 85.8 (87.8) | 86.3 | 90.5 / 87.4 (89.0) | 70 | 91.1 | 83.7 / 83.4 (83.6) |
| SR-STE (Zero Dense) | 1:8(FF) + 1:8(QKV) | 73.2 | 13.5 | 76.3 | 76.4 | 89.0 / 84.6 (86.8) | 83.2 | 89.5 / 85.9 (87.7) | 63.9 | 87 | 84.3 / 84.2 (84.2) |
| SR-STE (10K Dense) | 1:8(FF) + 1:8(QKV) | 74.2 | 16.1 | 77.7 | 77.6 | 88.5 / 84.1 (86.3) | 82.9 | 89.9 / 86.3 (88.1) | 66.4 | 88.8 | 84.4 / 84.2 (84.3) |
| MDGF-Stepwise (10K Dense) | 1:8(FF) + 1:8(QKV) | **79.5** | **33** | **82.3** | **82.3** | **91.3 / 87.7 (89.5)** | **89.2** | **91.0 / 88.0 (89.5)** | **74.4** | **91.1** | **84.5 / 84.8 (84.6)** |
| MDGF-Geometric (Zero Dense) | 1:8(FF) + 1:8(QKV) | 75.5 | 22.1 | 78.6 | 78.7 | 90.5 / 86.8 (88.6) | 83.4 | 90.0 / 86.5 (88.2) | 67.9 | 88.2 | 84.2 / 84.2 (84.2) |
| MDGF-Geometric (10K Dense) | 1:8(FF) + 1:8(QKV) | 75.8 | 19.5 | 79.4 | 79.6 | 89.4 / 85.3 (87.3) | 84.5 | 90.2 / 86.8 (88.5) | 70.4 | 89.8 | 83.3 / 83.0 (83.2) |
| SR-STE (Zero Dense) | 1:8(FF) + 1:4(QKV) | 75.1 | 15 | 78.4 | 79 | 90.5 / 86.8 (88.6) | 84.2 | 90.1 / 86.6 (88.4) | 67.9 | 88.4 | **86.2 / 86.1 (86.2)** |
| SR-STE (10K Dense) | 1:8(FF) + 1:4(QKV) | 78 | 24.5 | 81.2 | 81.6 | 91.1 / 87.7 (89.4) | 87.1 | 90.6 / 87.3 (89.0) | 72.2 | 90.9 | 85.8 / 85.8 (85.8) |
| MDGF-Stepwise (10K Dense) | 1:8(FF) + 1:4(QKV) | **80.3** | **36.4** | **83.2** | **83.4** | 90.9 / 87.3 (89.1) | **90.3** | **91.3 / 88.3 (89.8)** | **74.7** | 90.9 | 85.2 / 85.0 (85.1) |
| MDGF-Geometric (Zero Dense) | 1:8(FF) + 1:4(QKV) | 76.8 | 20.2 | 80.5 | 80.8 | **91.3 / 87.7 (89.5)** | 85.4 | 90.3 / 87.0 (88.6) | 70.8 | 90.4 | 84.9 / 84.9 (84.9) |
| MDGF-Geometric (10K Dense) | 1:8(FF) + 1:4(QKV) | 78.9 | 27.7 | 82.4 | 82.4 | **91.3 / 87.7 (89.5)** | 88.8 | 91.0 / 88.1 (89.6) | 74.4 | **91.3** | 84.5 / 84.5 (84.5) |

Here is an itemized list of nine tasks used in the GLUE dataset, along with brief descriptions of each:

- **CoLA (Corpus of Linguistic Acceptability)**: Classify whether a given sentence is grammatically acceptable or not.

- **MNLI (Multi-Genre Natural Language Inference)**: Classify the relationship between a given premise and hypothesis as entailment, contradiction, or neutral. We use the standard test set, for which we obtained private labels from the authors, and evaluate on both the matched (in-domain) and mismatched (cross-domain) sections.

- **MRPC (Microsoft Research Paraphrase Corpus)**: Determine whether a pair of sentences express the same meaning or not.

- **QNLI (Question-answering Natural Language Inference)**: Determine whether a given question can be answered correctly using a given sentence.

- **QQP (Quora Question Pairs)**: Determine whether a pair of questions from Quora are semantically equivalent or not.

- **RTE (Recognizing Textual Entailment)**: Classify the relationship between a given premise and hypothesis as entailment or not.

- **SST-2 (Stanford Sentiment Treebank)**: Determine the sentiment of a given sentence as either positive or negative.

- **STS-B (Semantic Textual Similarity Benchmark)**: Calculate the similarity score between two sentences on a scale from 0 to 5.

These tasks cover various aspects of language understanding, including sentence acceptability, sentiment analysis, paraphrase detection, textual similarity, natural language inference, question-answering, and co-reference resolution.

Figure 7 shows the accuracy vs. fine-tuneing step curve for each of the 9 benchmarks of GLUE.

# C. Detailed Experimental Settings

## C.1. Datasets

### C.1.1. ImageNet-1K

ImageNet-1K [81] is a large-scale image classification task, known as one of the most challenging image classification benchmarks. It consists of more than 1.2 million training images and 50K validation images with a size of 224x224 pixels, each with 3 channels. Each image is labeled as one of the 1K classes. We use this dataset for studies in Section 4.1 of the main paper. For ViT and SwinV2 experiments, we use a patch size of 16. This converts the 224x224 pixel image into an input of sequence length $224/16 * 224/16 = 196$.

**Evaluation metrics.** All reported results follow standard Top-1 validation accuracy.

### C.1.2. CIFAR10

CIFAR-10 [74] is a smaller-scale image classification dataset consisting of 10 classes. Each class has 6000 color images of 32x32 pixels in size.

**Evaluation metrics.** All reported results to follow standard Top-1 accuracy.

### C.1.3. GLUE

The General Language Understanding Evaluation (GLUE) [84] benchmark is a collection of resources for training, evaluating, and analyzing natural language understanding systems. GLUE consists of: A benchmark of nine sentence- or sentence-pair language understanding tasks built on established existing datasets and selected to cover a diverse range of dataset sizes, text genres, and degrees of difficulty, Table 12 shows the overall score for each sparsity target using different sparsification methods.

**Evaluation metrics.** All reported results in the main paper use the overall average score.

### C.1.4. WMT

WMT-17 (English-German) [85] is a key benchmark in machine translation research. They hold several translation datasets across different languages. The training set consists of about 4.5 million bilingual sentence pairs from WMT 2014.

**Evaluation metrics.** We calculate accuracy by comparing the translated output to the correct translation in the validation datasets.

## C.2. Hyperparameters for Different Models

### C.2.1. Image Classification → Vision Transformers (`ViT`)

We train the ViT-Base model on ImageNet-1k with hyperparameters presented in Table 13. We follow the hyperparameter setting in [80] for all ViT experiments. We also use the same hyperparameters to train ViT-Tiny model ( 3 layers, 3 attention head per layer, Embedding dimension: 192) on CIFAR-10 for initial experiments in Section 3.2 for analysing the trends of weights, gradients and optimizer moments and comparing those with SR-STE.

The detailed list of all hyperparameters can be found at hyperparaters.yaml. For ViT-Base, the training phase takes $\approx 44$ hours on 16 - A100 GPUs.

Figure 6 shows the Top-1 and Top-5 accuracy trends for training ViT to various sparsity targets with different sparsification techniques. We observe generally, MDGF and SDGF are better than SR-STE, especially for high-sparsity targets.

Table 13: Hyperparameters used for training ViT on ImageNet-1K.

| | |
|---|---|
| Batch Size | 256 |
| Training Epoches | 350 |
| Learning Rate | 1e-3 |
| LR Warmup Epoches | 15 |
| LR Decay schedular | Cosine |
| Decay Rate | 0.1 |
| Decay Epoches | 100 |
| Optimizer | AdamW |
| Optimizer coefs | beta1 = 0.9, beta2 = 0.999 |

### C.2.2. Image Classification → Swin Transformer V2 (`SwinV2`)

We train the SwinV2-Base model on imagenet-1k with hyperparameters presented in Table 14. We follow the hyperparameter setting in [79] for all SwinV2 experiments.

Table 14: Hyperparameters used for training SwinV2 on ImageNet-1K.

| | |
|---|---|
| Batch Size | 128 |
| Training Epoches | 350 |
| Learning Rate | 1e-3 |
| LR Warmup Epoches | 20 |
| LR Decay schedular | Cosine |
| Decay Rate | 0.1 |
| Decay Epoches | 30 |
| Optimizer | AdamW |
| Optimizer coefs | beta1 = 0.9, beta2 = 0.999 |

The detailed model configuration is the same as present in the original Microsoft research GitHub repo, SwinV2-base.yaml The detailed list of all hyperparameters was taken from config.yaml. For SwinV2-Base, the training phase takes ≈ 54 hours on 16 - A100 GPUs.

### C.2.3. Language Understanding → `T5X`

We train the T5X-Base model on GLUE dataset with hyperparameters presented in Table 15. We follow the hyperparameter setting in [1] for all T5X training experiments.

The detailed model configuration is the same as present in the original Google research GitHub repo, T5X model T5X-Base's training phase takes ≈ 22 hours on 8×Google Cloud TPUv3 cores.

Table 15: Hyperparameters used for training T5X on GLUE.

| | |
|---|---|
| Batch Size | 128 |
| Training Steps | 100k |
| Learning Rate | 1e-3 |
| LR Warmup Steps | 1000 |
| LR Decay schedular | Constant |
| Optimizer | AdamW |
| Optimizer coefs | beta1 = 0.9, beta2 = 0.999 |

### C.2.4. Language Translation Model → `Enc-Dec`

We train an encoder-decoder-based model on WMT-17 with hyperparameters presented in Table 16. The model is inspired by the attention paper [86]. We follow the hyperparameter setting in [89] to train all models. The training phase takes ≈ 8 hours on 32 - Google Cloud TPU v3 cores.

Table 16: Model configurations and hyperparameters for training model on WMT.

| | |
|---|---:|
| Number of Encoder Layers | 6 |
| Number of Decoder Layer | 6 |
| Hidden Dimension Size | 1024 |
| Feed-Forward Dimension Size | 4096 |
| Number of Attention Heads | 16 |
| Max Sequence Length | 256 |
| Training Dataset | WMT-17 |
| Testing Dataset | WMT-14 |
| Batch Size | 512 |
| Training Steps | 200K |
| Learning Rate | 0.0625 |
| LR Warmup Steps | 1000 |
| Decay Factor | 0.5 |
| Optimizer | Adam |
| Optimizer coefs | beta1 = 0.9, beta2 = 0.92 |

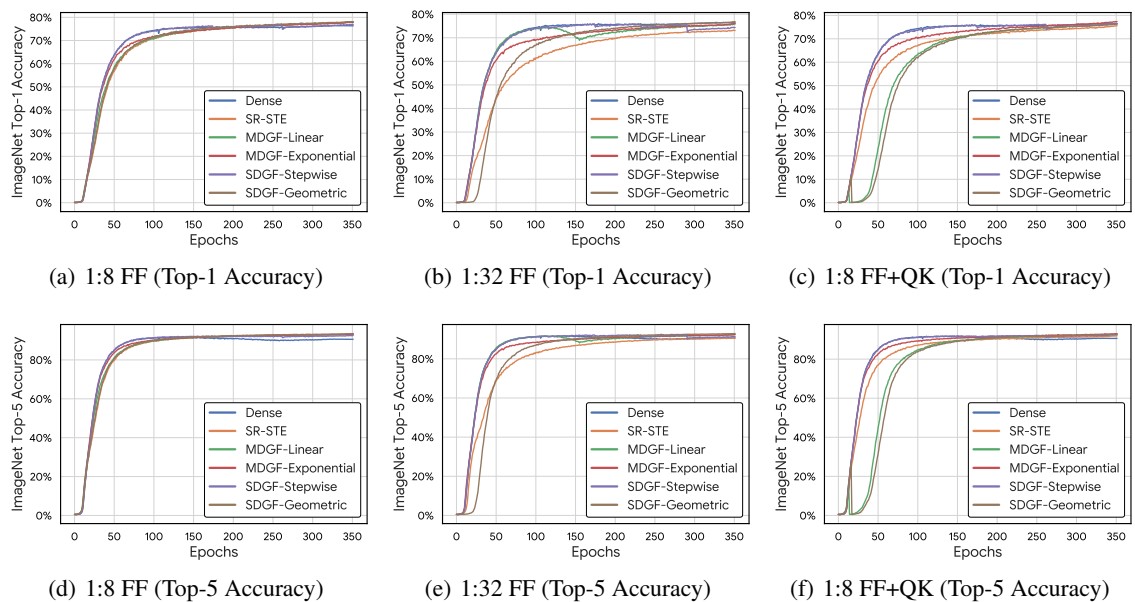

(a) 1:8 FF (Top-1 Accuracy)  (b) 1:32 FF (Top-1 Accuracy)  (c) 1:8 FF+QK (Top-1 Accuracy)

(d) 1:8 FF (Top-5 Accuracy)  (e) 1:32 FF (Top-5 Accuracy)  (f) 1:8 FF+QK (Top-5 Accuracy)

Fig. 6: Training Epochs vs Accuracy graph for different sparsity targets. We train `ViT-Base` on ImageNet-1K.

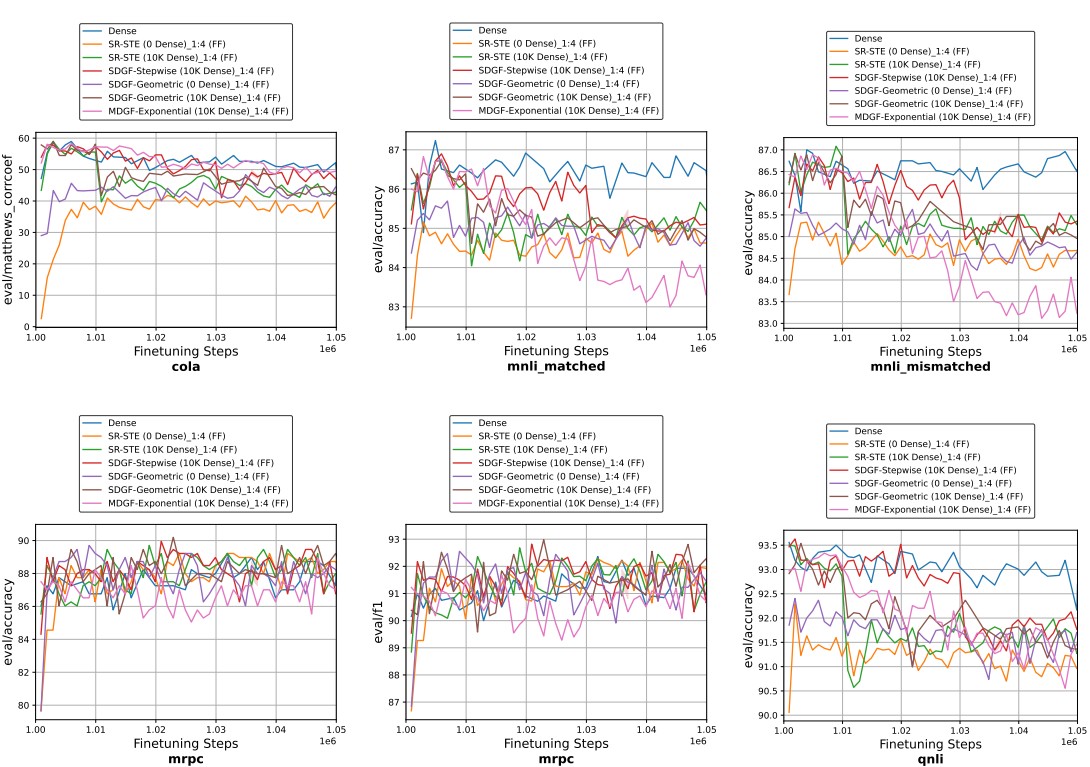

Fig. 7: Per-task evaluations for `T5X-Base` model finetuned on the GLUE dataset for 50 K steps.

# D. FLOPS Calculation

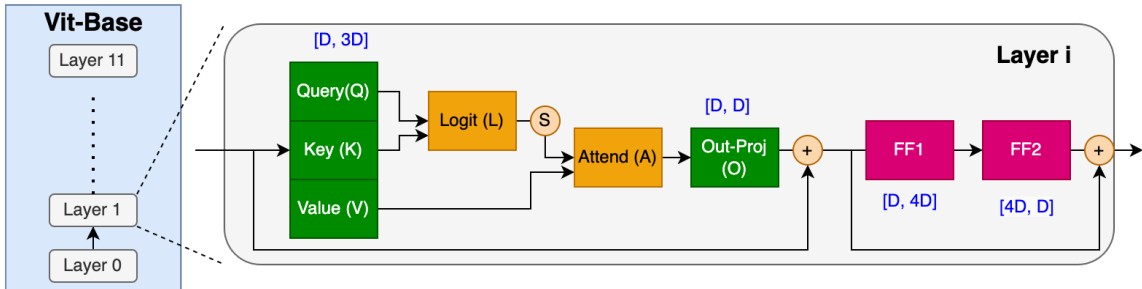

Fig. 8: Operations for ViT base model. For sake of brevity, we only include the operators that take significant runtime. Parameter dimensions are mentioned in blue text near the corresponding operators.

Figure 8 shows various operators in ViT base model. The breakdown of flops, Table 17, shows that FF accounts for majority of the FLOPS and thus would be our main avenue of sparsification.

| FLOPS (G) | Q/K/V/O | L/A | FF1/FF2 |
|---|---|---|---|
| Dense | 2.77 | 0.7 | 11.1 |

Table 17: Operator wise FLOPS breakdown for ViT-base.

We calculate the total number of flops for the model as follows.

$$FLOPS_{tot} = FLOPS_{SA} + FLOPS_{FF} * S_{FF}$$
$$FLOPS_{SA} = FLOPS_Q + FLOPS_K + FLOPS_V + FLOPS_L + FLOPS_A + FLOPS_O$$
$$FLOPS_{FF} = FLOPS_{FF1} + FLOPS_{FF2}$$

$FLOPS_{SA}$ is number of flops in self-attention layers which consists of QKV generation, 2 einsums (Logit and Attend) and output projection(O).

$FLOPS_{FF}$ is number of flops of the 2 feed-forward layers.

Using these equations, We list the total FLOPS of ViT-base for various sparsity targets in Table 18.

| Sparsity : $S_{FF}$ | $FLOPS_{SA}$ | $FLOPS_{FF}$ | $FLOPS_{tot}$ |
|---|---|---|---|
| Dense : 1.0 | 12.51 | 22.19 | 34.71 |
| 2:4 (FF) : 0.5 | 12.51 | 11.1 | 23.61 |
| 1:4 (FF) : 0.25 | 12.51 | 5.55 | 18.06 |
| 1:8 (FF) : 0.125 | 12.51 | 2.77 | 15.29 |
| 1:16 (FF) : 0.0625 | 12.51 | 1.39 | 13.90 |
| 1:32 (FF) : 0.03125 | 12.51 | 0.69 | 13.20 |
| 1:128 (FF) : 0.0078125 | 12.51 | 0.17 | 12.69 |

Table 18: FLOPS(G) calculation for various level of sparsity in ViT-Base.

# E. Sparse Matmuls speedup on Real Hardware.

We have conducted additional experiments showcasing the benefits of other N:M sparsity forms in performing sparse matrix multiplications on hardware without compute support for N:M (!= 2:4) acceleration.

We performed these experiments using real hardware, specifically V100, A100, and GH200. We used cupy [90] library along with the spmatrix.dot [91] function for sparse computation. We measured the run-time (after a few iterations of warm-up) of different feedforward kernels of ViT.

Table 19 shows the speedup of running VIT FFN kernels on different hardware. As shown, different forms of N:M sparsity offer speedup over default 2:4 structured sparsity (up to 8.62 on V100, 5.87 on A100 and 4.16 on H100).

These results further support our claim on the benefits of N:M sparsity variants in delivering performance. Note that the benefits of non-2:4 kernels can be primarily attributed to memory savings, reduction in the data communication, and customized cusparse API. We also included nsys profiler logs (Figure 9) to ensure the benefits originate from both memory savings and customized cusparse kernel.

Table 19: Average Speedup across FF1 and FF2 compared to 2:4 for different sparsity levels.

| Hardware | 2:4 | 1:4 | 1:8 | 1:16 | 1:32 | 1:128 |
|---|---|---|---|---|---|---|
| V100 | 1.0 | 1.83 | 3.05 | 5.25 | 7.64 | 8.62 |
| A100 | 1.0 | 1.85 | 2.96 | 4.01 | 4.82 | 5.87 |
| GH200 | 1.0 | 1.80 | 2.58 | 3.22 | 3.65 | 4.16 |

Fig. 9: Nsys profiler log for running N:M hardware on A100.

