# OpenReview forum: "Progressive Gradient Flow for Robust N:M Sparsity Training in Transformers"
_CPAL.cc/2025/Proceedings_Track — CPAL 2025 (Proceedings Track) Oral_

### Official Review · Reviewer_i5wz · 2025-01-10

**Rating:** 7
**Confidence:** 3

**Review:**

The work studies the training of N:M sparse deep networks. A network is N:M sparse when, if we divide its weight matrices into several blocks each with M elements, at most N of them are nonzero in each block. Such networks are faster to infer due to specialized hardware, but are difficult to train, and pruning the weights after training may lead to a decay in performance. The paper identifies an issue with the previous SoTA method SR-STE, which has large gradient and second-moment variance when training on typical tasks with Adam, and then proposes two fixes MdGf and SdGf which anneal the sparsity to a small ratio or 0 (continuously, as in approximate sparsity, for the first method, and discretely for the second method) throughout training. The paper then conducts experiments on MdGF and SgDf where they perform favorably to the SoTA SR-STE.

The paper has several positive points, which push in the direction of acceptance:
1) The methods MdGf and SdGf are quite simple conceptually to describe, if not to implement; it is nice that such methods can perform well in practice. Each method has two alternatives described in the paper, which amount to choosing different annealing schedules: MdGf-Linear, MdGf-Exponential, SdGf-Stepwise, SdGf-Geometric.
2) The experimental results are overall good; at least one of the four above methods is SoTA on the attempted tasks, and higher sparsity ratios are more tolerable.

However:
1) It is not clear which of the four methods should be used in any practical scenario. It seems that they are all effective in different scenarios, but to distinguish between them needs empiricla trial and error. Guidance on this point may improve the paper and make the methods easier to use.
2) Furthermore, even though the methods are simple to describe, the latter two methods may be difficult to come up with, and intuition about how the annealing procedures are found (and how the hyperparameters are chosen) would improve the paper.
3) It seems like no autoregressive language models are trained, despite all other types of models being trained. Do the methods work for autoregressive language models? This is a high-impact application and if the method enables higher sparsity it can be very useful.

---

### Official Review · Reviewer_1ud2 · 2025-01-10
**Reviews**

**Rating:** 6
**Confidence:** 3

**Review:**

The paper proposes a novel Progressive Gradient Flow (PGF) method for sparse training, specifically targeting N:M structured sparsity in deep neural networks. The authors introduce a gradient decay mechanism aimed at reducing gradient noise in high-sparsity regimes. Empirical results on image and language tasks, as well as hardware performance benchmarks, demonstrate significant improvements in model accuracy and computational efficiency compared to existing methods.

**Strengths:**

1. The paper addresses a critical issue of gradient noise in high-sparsity training.

2. The proposed decaying gradient flow technique demonstrates promising results, with accuracy improvements of up to 5% and significant FLOP reductions.

3. The empirical evaluation covers multiple datasets and hardware platforms, enhancing the paper's practical relevance.

**Questions:**

1. Why improving gradient flow reduces gradient noise.

2. What do \(\beta\) and \(j\) represent in line 115?

3. Why is \(\tilde{\mathcal{W}}\) sparse? \(\Phi(\mathcal{W}, N, M, j)\) is a binary mask, but \([\Phi(\mathcal{W}, N, M, j) + \mathcal{D}(j)(1 - \Phi(\mathcal{W}, N, M, j))]\) is not a sparse matrix.

5. It is also unclear why your method can reduce gradient noise, and the motivation for proposing this method is not clear to me.


**Weakness:**

Unclear Motivation and Theoretical Justification: The paper does not clearly explain why decaying gradient flow reduces gradient noise. The link between restricted gradient flow and noise reduction needs to be better justified with theoretical reasoning or references to existing variance reduction techniques.

---

### Official Review · Reviewer_xk4Z · 2025-01-11

**Rating:** 7
**Confidence:** 4

**Review:**

This paper suggests that current N:M structure training is less effective at high sparsity levels, due to the noise in gradients.
To address this issue, this paper proposes the decaying-based training recipes, “Mask Decay Gradient Flow” (MDGF), and “Structure Decay Gradient Flow” (SDGF). Specifically, instead of using the fixed mask from magnitude-based pruning, MDGF applies floating-point masks which gradually diminish from 1 to 0, and SDGF gradually alters the N:M sparsity to the target sparsity level.
This paper conducts extensive experiments with multiple network architectures to demonstrate the effectiveness of the proposed training recipes for N:M sparsity at high-sparsity level.

The paper is well-crafted, and the improvement achieved at high levels of sparsity, compared to the state-of-the-art SR-STE, is remarkable.

Here are some weaknesses that can be further improved:
- It's better to present the validation loss/accuracy during training in Figure 3 to verify that the variance of second moment is correlated with the convergence speed. Currently, this paper only cites previous works (Line 172) and use descriptive sentences ("When observing the final validation accuracy of the two runs, we confirm ...") to support this claim, which is vague.
Without the validation loss/accuracy, it's difficult to draw the conclusion that the noisy gradient flow is the reason for the performance drop at high-level sparsity in Section 4.

- The presentation in some figures can be further improved. For example, in Figure 1 (b), should the output be $\tilde{W_t}$ for sparse weights, as similar to Figure 2(a)?
- It looks like there is no limitation discussion in Section 6, although the title is "Limitation and Future work".

---

### Meta-Review · Area_Chair_jiZw · 2025-02-02

**Recommendation:** Accept (Poster)
**Confidence:** 4

**Metareview:**

This paper proposes novel training recipes to achieve N:M structured sparsity in deep networks. The key problem addressed is the issue of gradient noise in high-sparsity training, which limits accuracy of models trained using standard approaches. The authors propose various training strategies based on replacing binary weight masks with a floating point masks that continuously decay from 1 to 0, which is conjectured to reduce noise in the gradients. The training strategies are demonstrated on a variety of practical image and language tasks.

Reviewers agree the paper is well-motivated, practically relevant, and were mostly satisfied with the empirical demonstrations. Several reviewers were impressed with the improvements in model accuracy and computational efficiency over the previous state-of-the-art.  A few common weaknesses were identified: a lack of theoretical justification of why decaying gradient flow reduces gradient noise, a limited scope to the experiments (e.g., autoregressive LLMs), and some issues with presentation of the results. In their rebuttals, the authors largely agreed with the reviewers on these points.

Overall, despite some minor weaknesses, I believe the paper makes a valuable contribution to the field and should be accepted. I encourage the authors to incorporate the reviewers' feedback into the final version of the paper, especially regarding issues of presentation with the figures, providing more intuition for why their proposed training methods reduce gradient noise, and fleshing out the limitations section in the paper.

---

### Decision · Program_Chairs · 2025-02-11

Accept (Oral)